# *Dig-up Primers*: A Pipeline for Identification of Polymorphic Microsatellites Loci within Assemblies of Related Species

**DOI:** 10.3390/ijms25063169

**Published:** 2024-03-09

**Authors:** Ante Turudić, Zlatko Liber, Martina Grdiša, Jernej Jakše, Filip Varga, Igor Poljak, Zlatko Šatović

**Affiliations:** 1Centre of Excellence for Biodiversity and Molecular Plant Breeding (CoE CroP-BioDiv), Svetošimunska Cesta 25, 10000 Zagreb, Croatia; aturudic@agr.hr (A.T.); mgrdisa@agr.hr (M.G.); zsatovic@agr.hr (Z.Š.); 2Faculty of Agriculture, University of Zagreb, Svetošimunska Cesta 25, 10000 Zagreb, Croatia; zlatko.liber@biol.pmf.hr; 3Faculty of Science, University of Zagreb, Marulićev trg 9a, 10000 Zagreb, Croatia; 4Biotechnical Faculty, University of Ljubljana, Jamnikarjeva 101, 1000 Ljubljana, Slovenia; jernej.jakse@bf.uni-lj.si; 5Faculty of Forestry and Wood Technology, University of Zagreb, Svetošimunska Cesta 23, 10000 Zagreb, Croatia; ipoljak@sumfak.hr

**Keywords:** SSRs, NGS, SSR primer design, in silico PCR

## Abstract

Simple sequence repeats (SSRs) have become one of the most popular molecular markers and are used in numerous fields, including conservation genetics, population genetic studies, and genetic mapping. Advances in next-generation sequencing technology and the growing amount of genomic data are driving the development of bioinformatics tools for SSR marker design. These tools work with different combinations of input data, which can be raw reads or assemblies, and with one or more input datasets. We present here a new strategy and implementation of a simple standalone pipeline that utilizes more than one assembly for the in silico design of PCR primers for microsatellite loci in more than one species. Primers are tested in silico to determine if they are polymorphic, eliminating the need to test time-consuming cross-species amplification in the laboratory. The end result is a set of markers that are in silico polymorphic in all analyzed species and have great potential for the identification of interspecies hybrids. The efficiency of the tool is demonstrated using two examples at different taxonomic levels and with different numbers of input assemblies to generate promising, high-quality SSR markers.

## 1. Introduction

Simple sequence repeats (SSRs), also known as microsatellites, are a part of the genome that contains a large number of copies of a particular motif consisting of one to six nucleotides [1,2,3]. In the genomes of eukaryotes and prokaryotes, microsatellite loci are found in both non-coding and coding regions [4,5,6,7]. Because they are codominant, multiallelic, and highly repeatable, they are the most widely used molecular markers in numerous fields, including ecology, conservation genetics, population genetic studies, paternity determination, genetic mapping, and identification of lines and varieties in breeding programs [8,9,10,11,12]. The characteristics that give microsatellites an advantage over other molecular markers are their codominance and their high informativeness, which results from their extremely high mutation rate compared to the rest of the genome, ranging between 10^−3^ and 10^−6^ per locus and generation [13,14,15]. There are several mechanisms responsible for such a high mutation rate, with replication slippage emerging as the most important [16]. The high mutation rate is also the major drawback of microsatellites, as it can lead to back mutations that promote homoplasy, especially when different species are studied with microsatellite markers developed for only one of them. Due to this problem and the high variability of the flanking regions, microsatellite markers have usually had to be developed anew for each species [17,18].

SSRs have been used in genetic studies for several decades, and in recent years an increasing number of new methods based on next-generation sequencing (NGS) techniques (e.g., Restriction-site associated DNA sequencing; RADseq) have been offered to replace them. There are numerous publications comparing SSRs with these new techniques (e.g., [18,19,20]). The general conclusion is that, despite the method used, the final results are very similar, with a slightly higher resolution from next-generation sequencing methods, while SSRs are cheaper and easier to use. The conclusion is that SSRs are still competitive and that the choice of markers depends on the objectives of the study and the resources available.

The traditional laboratory approach to developing SSR markers involves the construction of SSR libraries. The first step in this procedure is the digestion of DNA with a large number of restriction enzymes and subsequent ligation of linkers with known sequences to the resulting restriction fragments, which are hybridized in the next step with synthetic DNA probes containing SSR sequences. The SSR-enriched fragments are then amplified by Polymerase chain reaction (PCR), cloned, and sequenced. The final step is to construct PCR primers and test the SSR markers in test populations [2,9,21]. NGS sequencing has fundamentally changed the process of searching for SSR markers and made it more efficient by finding hundreds of SSR markers cheaper and faster [22].

In general, SSR markers are species-specific, so primers are usually designed anew for each species [23]. Sometimes flanking regions are conserved across taxa, allowing cross-species amplification of SSR markers with primers designed for other species of the same genus or even family [24]. However, cross-species amplification in related species often results in null alleles, monomorphic loci, and lower allelic richness compared to the target species [25,26], leading to biased estimation of allelic and genotypic frequencies and underestimation of heterozygosity [8]. If in silico cross-amplification were to be performed on genome assemblies from two or more closely related species, any bias that can be caused by multi-species analysis could be overcome with bioinformatic approaches that share assemblies from the species of interest. This approach could also completely eliminate the tedious and time-consuming cross-species amplification tests in the laboratory.

In phylogeny, phylogeography, population, and conservation genetics, there is great interest in determining interspecific hybridization and introgression in natural systems, though doing so is challenging [27]. The accurate genetic determination of hybrid individuals within populations is particularly important in the study of plant species where hybridization occurs frequently [28,29,30,31]. Microsatellite markers developed in silico through various NGS and bioinformatics approaches [18], which have been shown to represent the homologous microsatellite loci and exhibit a high degree of polymorphism in hybridized species, have great potential for the identification of hybrid individuals. Advances in NGS sequencing technologies such as the sequencing of 99.9% accurate reads longer than a thousand base pairs [32], new statistical tools for hybrid assessment (e.g., [33,34,35,36,37,38,39,40,41]), and a large amount of public genomic data enable the development of bioinformatics tools for the discovery of promising SSR primers for the study of various biological phenomena, including hybridization, in numerous plant species.

There are many bioinformatics tools for the identification and development of SSR markers based on NGS data (such as QDD [42], CandiSSR [43], Mimi [44], IDSSR [45], 3GMAT [46], SSR2Marker [47]), including databases that store SSR marker data of a single or several related species (such as citSATdb [48], LegumeSSRdb [49], PmiRNASSRdb [50], MMdb [46]). Since the abovementioned tools work with a dataset (raw reads or assembly) of a single species, the aim of this study was to develop a bioinformatics pipeline that utilizes the genome assemblies of related species to find microsatellite markers that amplify in silico and are polymorphic in all species of interest. As a model species, we used one assembly of common ash (*Fraxinus excelsior* L.) and three assemblies of narrow-leaved ash (*Fraxinus angustifolia* Vahl, corresponding to the following three subspecies: subsp. *angustifolia* (Vahl) Wesm., subsp. *oxycarpa* (M.Bieb. ex Willd.) Franco & Rocha Afonso, and subsp. *syriaca* (Boiss.) Yalt.) available at the National Center for Biotechnology Information (https://www.ncbi.nlm.nih.gov/, accessed on 22 October 2023). The *Fraxinus* species were selected because they hybridize readily and there is a growing body of research examining the ecology of hybridization and its effects on ecosystems [51,52].

## 2. Results

We performed two identifications of SSR markers based on genome assemblies of species of the genus Fraxinus. One was between two species (*Fraxinus excelsior* and *F. angustifolia*) and the second was among subspecies of *F. angustifolia*: subsp. *angustifolia*, subsp. *oxycarpa* and subsp. *syriaca*. For identification, we used the pipeline *Dig-up Primers* developed for the study and ran it with default parameters (see Section 4.3). We ran the program on a ThinkPad P14s laptop (AMD Ryzen 7 PRO 5850U processor, 32 GB RAM) with 16 threads.

### 2.1. Joint SSR Marker Identification in Two Species

The genome assembly of *F. excelsior* was on the chromosome level and had a small number of contigs (421), which made it the most suitable base for SSR mining. That assembly was used as Assembly A (Figure 1), on which the first four pipeline steps were performed. For *F. angustifolia*, we used the genome of subspecies subsp. *oxycarpa*, since it was the longest (714.3 Mb) genome assembled at scaffold level (Assembly B in Figure 1).

In Assembly A, we identified 28,403 SSRs, of which 25,514 SSR regions were eligible for further review. Of the eligible SSR regions, 13,804 were unsuitable for primers with the indicated parameters, and 5980 were discarded because they were regions of low complexity or close to coding regions. In silico PCR was performed with 7824 primer pairs. Nine hundred and ten of the resulting SSR markers were amplified only once in both assemblies and were polymorphic (Table 1). The total execution time was 40 min and 9 s. The step that took the longest was the low-complexity check (step 3), which took 20 min and 26 s, followed by in silico PCR (step 4), which took 13 min and 16 s. These two steps took more than 80% of the total execution time.

### 2.2. Joint SSR Marker Identification in Three Subspecies

The three available genome assemblies of *F. angustifolia* subspecies were subsp. *oxycarpa* (Assembly A), subsp. *syriaca* (Assembly B), and subsp. *angustifolia* (Assembly C). The pipeline started with 18,793 identified SSRs, from which 12,618 regions were further screened. No suitable primers were found in 7643 regions, and 3069 regions failed the low-complexity check. After amplification in silico with 4574 primer pairs, 154 SSR markers were obtained that were amplified only once in each genome and were found to be polymorphic (Table 2). The total execution time was 24 min and 57 s. As in the previous case, the longest steps were the low-complexity check (10 min and 6 s) and in silico PCR (10 min and 33 s).

## 3. Discussion

In the present work, a pipeline was developed to identify polymorphic SSRs in related species. The pipeline was implemented using standard tools and is very easy to use, requiring only the input assemblies to be specified. The workflow starts with SSR identification. The following steps include the analysis of SSR regions, the design of SSR primers, and the low-complexity check to discard regions that do not fulfill the specified criteria. The final steps include in silico PCR on two (or more) assemblies, followed by the acquisition of final results containing only markers that were amplified exactly once in each assembly and that are polymorphic between the analyzed species. The final list includes SSR and primer sequences and their positions in all analyzed assemblies. The result of this approach is the rapid and simultaneous detection of easily generated alleles at the same polymorphic SSR loci in different taxa, eliminating the tedious and time-consuming cross-species amplification tests in the laboratory.

The criteria for filtering out unsuitable SSR regions were selected based on their known influence on the probability of successful amplification of SSR markers, whereby the values of all parameters can be adjusted. Since the number of SSR regions is usually large, it is advisable to set strict filter values. In general, the criteria can be categorized into three groups: (1) SSR structure (i.e., number of nucleotides in the repeating unit; maximum length), (2) SSR region (i.e., SSR is not located near other SSRs, the contig end, or the coding region; SSR is not located in a region of lower complexity), and (3) SSR marker polymorphism (i.e., SSR is amplified only once per assembly and shows length polymorphism between assemblies).

To evaluate its efficiency, the pipeline was used to identify SSRs on assemblies of species of the genus *Fraxinus* at two taxonomic levels and with a different number of input assemblies. Both calculations yielded enough promising SSR markers for further laboratory tests. The number of potentially useful SSR regions decreased to a similar extent in both runs, with the number of SSRs initially obtained depending on the size and level (e.g., chromosome, contig, or scaffold) of the assembly. As expected, in the subspecies *Fraxinus angustifolia* subsp. *oxycarpa* (subspecies level; Assembly A), which contained many short contigs, there was a significantly greater number of SSRs near the contig ends than in *Fraxinus excelsior* (species level; Assembly A), for which the genome assembly was available at the chromosome level. The computation time was short in both cases at about half an hour. Although the second calculation worked with three assemblies, it was completed faster because more SSRs were filtered out in the first two steps. The species of the genus *Fraxinus* in Europe grow in different habitats, which has led to a great diversity in their ecological and morphological characteristics [53]. *Fraxinus excelsior*, *F. angustifolia*, and *F. ornus* are all indigenous to Europe, of which the first two, the narrow-leaved ash and the common ash, are of great economic importance for forestry due to their high-quality wood [54]. From an ecological point of view, the common ash and the narrow-leaved ash prefer different habitats. However, in locations where the ecological niches of these two species overlap, their co-occurrence and thus hybridization is possible [52,55], which has been repeatedly confirmed in previous studies using both genetic and morphological methods [51,56]. Using SSR markers, two clearly separated genetic clusters of narrow-leaved ash and common ash have been identified in Europe, as well as a number of hybrid populations that differ in their proportion of the parental gene pools [52]. The hybridization of narrow-leaved and common ash has been studied in northern France [51,56], and hybrid populations also exist in Spain and along the Drava and Rhine rivers [57]. Nevertheless, gene introgression and hybridization between the two ash species have not been confirmed in all areas where sympatry exists [52], and further analyses are clearly needed.

## 4. Materials and Methods

### 4.1. Data Accessibility

The genome sequences of *Fraxinus* taxa were downloaded from the National Center for Biotechnology Information (NCBI) Genome Database (https://www.ncbi.nlm.nih.gov/genome/) (accessed on 22 October 2023). We acquired one available assembly of *Fraxinus* excelsior and three assemblies of *Fraxinus* angustifolia subspecies (subsp. *angustifolia*, subsp. *oxycarpa*, and subsp. *syriaca*). Details of the downloaded data can be found in Table 3.

### 4.2. Pipeline Availability and Requirements

The pipeline is called *Dig-up Primers*. It is implemented in the Python programming language and can be executed on all common operating systems. Python version 3 with the Biopython module was used [58]. External requirements are MISA [59], RepeatMasker [60], Primer3 [61], and BLAST [62]. The pipeline is optimized for the execution of external tools in multithreaded mode.

The homepage of the project is located at https://github.com/CroP-BioDiv/dig_up_primers (accessed on 2 February 2024).

### 4.3. Pipeline Process

*Dig-up Primers* was designed for mining SSR loci using more than one assembly, followed by designing suitable primers and testing their polymorphism in silico. The pipeline was developed as a local program and the input files were one or more genome assemblies in FASTA format.

The command line parameters control the operation of the program. All parameters, except the input files, have valid default values. The workflow consists of six steps: SSR identification, analysis of SSR regions, design of SSR primers, low-complexity check, in silico PCR, and acquisition of final results (Figure 1). In each step, the SSRs found, the region around them, and the markers designed for them were reviewed and filtered for features known to negatively affect the chances of SSR marker success. The first four steps are performed with the first assembly (A), as a marker must be found in each assembly to be included in the final result.

#### 4.3.1. SSR Identification

The mining of SSRs was carried out with the MISA software. The SSRs were only determined for the first specified assembly. It is possible to specify a motif length and a minimum number of repeats (default: 10 repeats for dinucleotide SSRs and 7 repeats for trinucleotide SSRs) as well as a maximum length of the SSR (default: 80 bp) in order to control the mining.

#### 4.3.2. SSR Region Analysis

The MISA output was processed to identify SSRs that were (a) simple (i.e., uninterrupted), (b) not near other SSRs, and (c) far enough away from the contig ends. There is a parameter that determines the minimum length of the regions between neighboring SSRs and between an SSR and the contig end (default: 200 bp).

#### 4.3.3. SSR Primer Design

The SSR regions were transferred to the Primer3 program for primer design. The properties of the designed primers were controlled by command line parameters and included the product size range (default: 100 to 250 bp), the number of results to be returned for a region (default: one), the minimum (default: 19 bp) and maximum size (default: 23 bp) of the primers, the minimum (default: 58 °C) and maximum melting temperature (default: 62 °C), the minimum (default: 40%) and maximum percentage of GC content (default: 60%), and the maximum length of a mononucleotide repeat (default: four repeats).

#### 4.3.4. Low-Complexity Check

The DNA segments between the designed primers were processed with the RepeatMasker program to check for regions of low complexity and proximity to coding regions. RepeatMasker is expected to flag SSRs as low-complexity regions, and likely some surrounding base pairs as well, since MISA has a more stringent strategy for identifying SSRs (e.g., in the case of a sequence ATGATGAT, MISA will flag ATGATG as a repeat, while RepeatMasker will flag ATGATGAT as a low-complexity region). A command line parameter controls how many surrounding base pairs are allowed (default: 5 bp). Segments with additional low-complexity regions were excluded from further processing, as were segments found near the coding region.

#### 4.3.5. In Silico PCR

SSR markers were amplified in silico in all input assemblies. Amplifications were performed using the BLAST program. The primers were located with BLAST using the short sequence search (task parameter blastn-short), whereby the identity was set to 100% and it was also checked whether the entire primer was found. The SSR marker was treated as amplified if the forward and reverse primers were close to each other and in the correct direction. The program parameter controls the maximum amplification length between primers (default: 1500 bp).

#### 4.3.6. Final Result

An SSR marker was selected if (a) it was amplified exactly once in each assembly, (b) all amplified regions contained SSRs with the same motif as the one for which the primers were designed, and (c) the differences in length of the amplified regions were exclusively caused by SSR length polymorphisms. The resulting SSR markers were collected together with all the useful information gathered in the previous steps.

## 5. Conclusions

*Dig-up Primers* is a newly developed bioinformatics tool that allows users to identify SSR markers that have been tested in silico for polymorphism between assemblies of related target species. The pipeline generates promising, high-quality SSR markers, eliminating the need for tedious and time-consuming cross-species amplification testing in the laboratory.

## Figures and Tables

**Figure 1 ijms-25-03169-f001:**
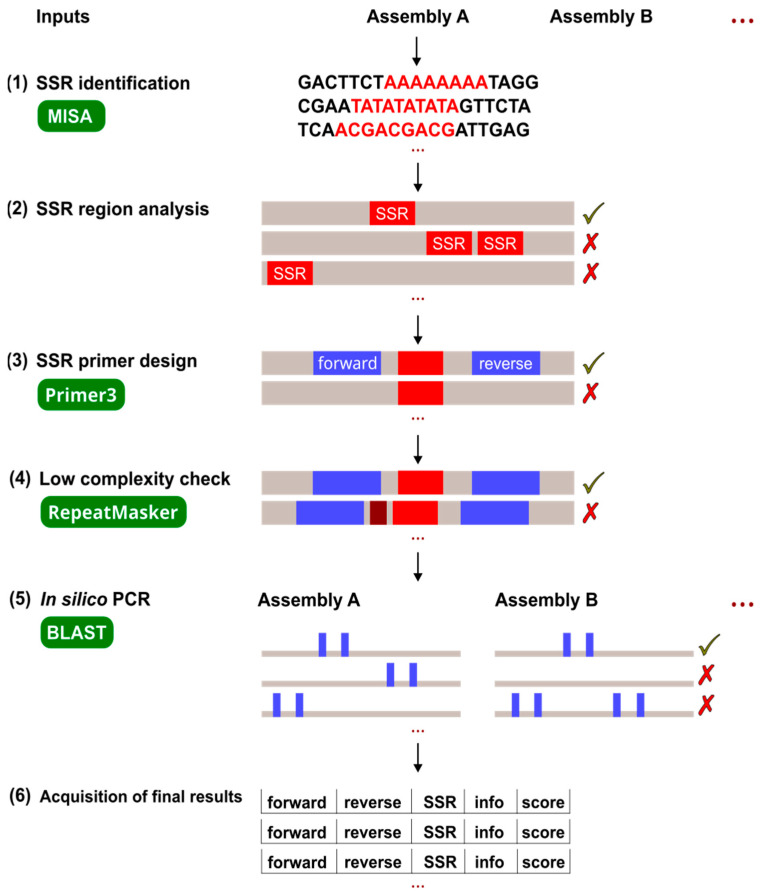
Flowchart of the pipeline process: (1) SSR identification with MISA, (2) analysis of SSR regions to find promising SSRs, (3) design of SSR primers with Primer3, (4) low complexity check with RepeatMasker, (5) in silico PCR with BLAST, and (6) acquisition of final results. The gray lines represent the DNA sequence, while the colored fragments represent the SSR sequence (red), primer sequence (blue), and additional low complexity region (brown). The intermediate results that are used for further processing are marked with a green check mark, while results that are unsuitable for further processing are marked with a red cross.

**Table 1 ijms-25-03169-t001:** Joint SSR marker identification in two species (*Fraxinus excelsior* and *F. angustifolia*): *Dig-up Primers* pipeline steps and the resulting number of SSRs included and excluded in each step.

Steps	Description	Included	Excluded
(1) SSR identification	Total no. of SSRs found	28,403	
(2) SSR region analysis	No. of composite SSRs		1137
	No. of long SSRs		162
	No. of SSRs close to another SSR		1584
	No. of SSRs close to the contig end		6
	No. of promising SSR regions	25,514	
(3) SSR primer design	No. of SSR regions without suitable primers		11,710
	No. of SSR regions with suitable primers	13,804	
(4) Low complexity check	No. of regions with low complexity		5902
	No. of regions close to a coding region		78
	No. of promising SSR markers	7824	
(5) In silico PCR	No. of markers amplified once in Assembly A	6777	
	No. of markers amplified once in Assembly B	1180	
(6) Acquisition of final results	Final no. of SSR markers	910	

**Table 2 ijms-25-03169-t002:** Joint SSR marker identification in three subspecies (*F. angustifolia* subsp. *oxycarpa*, subsp. *syriaca*, and subsp. *angustifolia*): *Dig-up Primers* pipeline steps and the resulting number of SSRs included and excluded in each step.

Steps	Description	Included	Excluded
(1) SSR identification	Total no. of SSRs found	18,793	
(2) SSR region analysis	No. of composite SSRs		397
	No. of long SSRs		4
	No. of SSRs close to another SSR		645
	No. of SSRs close to the contig end		5129
	No. of promising SSR regions	12,618	
(3) SSR primer design	No. of SSR regions without suitable primers		4975
	No. of SSR regions with suitable primers	7643	
(4) Low complexity check	No. of regions with low complexity		3021
	No. of regions close to a coding region		48
	No. of promising SSR markers	4574	
(5) In silico PCR	No. of markers amplified once in Assembly A	4403	
	No. of markers amplified once in Assembly B	477	
	No. of markers amplified once in Assembly C	771	
(6) Acquisition of final results	Final no. of SSR markers	154	

**Table 3 ijms-25-03169-t003:** Assemblies used in the analysis.

Species/Subspecies	Accession Number	Assembly Level	Assembly Size (Mb)	Number of Contigs
*Fraxinus excelsior*	GCA_019097785	Chromosome	806.5	421
*F. angustifolia* subsp. *angustifolia*	GCA_902829175	Contig	692.6	489,825
*F. angustifolia* subsp. *oxycarpa*	GCA_903798265	Scaffold	714.3	413,147
*F. angustifolia* subsp. *syriaca*	GCA_903798275	Scaffold	586.0	323,049

## Data Availability

The data that support the findings of this study are openly available in GenBank at the NCBI (https://www.ncbi.nlm.nih.gov (accessed on 22 October 2023)).

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
