# Peer review of "Dig-up Primers: A Pipeline for Identification of Polymorphic Microsatellites Loci within Assemblies of Related Species"

_ijms, 2024, doi:10.3390/ijms25063169_

Round 1

Reviewer 1 Report

Comments and Suggestions for Authors

This manuscript presents a new tool for in silico SSR mining and primer design. Its major advantage over existing tools is the co-design of primer pairs for closely related species, which is particularly pertinent when studying species that hybridize.
The manuscript is clearly, written and sound.

Major comments:
The first 4 steps of the pipeline are well designed and avoid pitfalls of designing primers in highly repetitive regions containing low complexity regions or several SSRs. I am a bit intrigued, however, by the in-silico PCR by using BLAST. I would like to have more information of how this step was parameterized apart from the length of the amplified region. Since primers are short sequences, it is difficult to find a correct E-value cut-off that detects all regions similar to the primers, and at the same time, do not include matches where there are relatively many mismatches at the 3’ of the primers. My feeling is that it would have been better to do first  low stringency BLAST, that detects many potential annealing sites, than further filter them based on the actual alignment between the contigs and the primer, where the number and type of mismatches and their position can be controlled (e.g. mismatches at 3’ can be more heavily penalized).  Another possibility would be to extract the amplicons obtained by low stringency BLAST, and force Primer3 to calculate primer scores between the extracted amplicon and the primers. This score can be used for discarding or keeping primer pairs.
A further potential improvement can be aligning the whole amplicons of different assemblies, to further ensure the homology between them. As I understand, at present, it is only the length of the amplicon and the match between the annealing sites and primers that are used to ensure homology.  This can be done easily by BLAST after soft-masking the SSRs. This suggestion is just a complement and should not replace the in-silico PCR, which is important to exclude primer pairs that match more than one region.
I think the pipeline as it is, is already quite robust. My suggestions are for a potential further development, to further increase the chance of getting useable markers, with as little lab test as possible.

At last, when users are faced with a few hundred potential primer pairs, they have to make a selection, since it is unlikely that will test all of them in the lab. One help could be to include the contig and the position of the primers on the contig, so that users can avoid SSR close to each other, and avoid genetic linkage.

Minor comments:
Lines 42-44: The problem of using cross-species microsatellites is not only the homoplasy, but variability of flanking regions. This should be mentioned here.

Author Response

R1: This manuscript presents a new tool for in silico SSR mining and primer design. Its major advantage over existing tools is the co-design of primer pairs for closely related species, which is particularly pertinent when studying species that hybridize. The manuscript is clearly, written and sound.

Response: Thank you very much for your kind words.

R1: Major comments:

The first 4 steps of the pipeline are well designed and avoid pitfalls of designing primers in highly repetitive regions containing low complexity regions or several SSRs. I am a bit intrigued, however, by the in-silico PCR by using BLAST. I would like to have more information of how this step was parameterized apart from the length of the amplified region. Since primers are short sequences, it is difficult to find a correct E-value cut-off that detects all regions similar to the primers, and at the same time, do not include matches where there are relatively many mismatches at the 3’ of the primers. My feeling is that it would have been better to do first  low stringency BLAST, that detects many potential annealing sites, than further filter them based on the actual alignment between the contigs and the primer, where the number and type of mismatches and their position can be controlled (e.g. mismatches at 3’ can be more heavily penalized).  Another possibility would be to extract the amplicons obtained by low stringency BLAST, and force Primer3 to calculate primer scores between the extracted amplicon and the primers. This score can be used for discarding or keeping primer pairs.

Response: We used BLAST very rigorously. We added a more detailed description to the chapter '4.3.5. In silico PCR' (l. 270-272).

R1: A further potential improvement can be aligning the whole amplicons of different assemblies, to further ensure the homology between them. As I understand, at present, it is only the length of the amplicon and the match between the annealing sites and primers that are used to ensure homology.  This can be done easily by BLAST after soft-masking the SSRs. This suggestion is just a complement and should not replace the in-silico PCR, which is important to exclude primer pairs that match more than one region.

I think the pipeline as it is, is already quite robust. My suggestions are for a potential further development, to further increase the chance of getting useable markers, with as little lab test as possible.

Response: Thank you for your suggestion. That would be a useful additional feature for future iterations of the pipeline.

R1: At last, when users are faced with a few hundred potential primer pairs, they have to make a selection, since it is unlikely that will test all of them in the lab. One help could be to include the contig and the position of the primers on the contig, so that users can avoid SSR close to each other, and avoid genetic linkage.

Response: The final results include the contig ID and SSR position on the contig so that users can make additional filtering before testing in the lab.

R1: Minor comments:

Lines 42-44: The problem of using cross-species microsatellites is not only the homoplasy, but variability of flanking regions. This should be mentioned here.

Response: Done. Lines 48-49.

Reviewer 2 Report

Comments and Suggestions for Authors

The authors developed a new strategy and implementation of a stand-alone pipeline that utilizes more than one assembly for the in silico design of PCR primers for SSR in more than one species. This method may eliminate the need to test time-consuming cross-species amplification in the laboratory. In addition, they displayed the efficiency of the tool using two examples at different taxonomic levels.

However, it seems the proposed method needs to be confirmed for different species. Also, similar to many methods developed by bioinformatics the efficiency of the new SSR may need to be confirmed in the lab for different utilities such as genetic diversity, mapping, etc. Or there are some statistical methods to test the work data?

Generally, the manuscript is well-written and the work is worth publishing after improving the discussion section and adding the conclusion section. 

Comments on the Quality of English Language

Minor editing of English language required

Author Response

R3: The authors developed a new strategy and implementation of a stand-alone pipeline that utilizes more than one assembly for the in silico design of PCR primers for SSR in more than one species. This method may eliminate the need to test time-consuming cross-species amplification in the laboratory. In addition, they displayed the efficiency of the tool using two examples at different taxonomic levels. However, it seems the proposed method needs to be confirmed for different species. Also, similar to many methods developed by bioinformatics the efficiency of the new SSR may need to be confirmed in the lab for different utilities such as genetic diversity, mapping, etc. Or there are some statistical methods to test the work data?

Response: During the development of the pipeline, we checked its efficiency using several suitable datasets we found in the NCBI, such as Nicotiana (N. tabacum, N. otophora, N. sylvestric, N. rustica), Malus (M. domestica, M. sylvestris), Allium (A. cepa, A. sativum) and Capsella (C. bursa-pastoris, C. rubella) datasets, and the results were as satisfactory as for the Fraxinus dataset. We are not aware of any statistical method that could be used to further test the markers. Since this method uses more than one assemly to test in-silico for marker polymorphism, we believe that the success rate will be satisfactory.

R3: Generally, the manuscript is well-written and the work is worth publishing after improving the discussion section and adding the conclusion section.

Response: We improved the descussion section (l. 158-160) and added the conslusion section (l. . 285-290).